# The Epidemiology of Deficiency of Vitamin B12 in Preschool Children in Turkey

**DOI:** 10.3390/medicina59101809

**Published:** 2023-10-11

**Authors:** Yusuf Elgormus, Omer Okuyan, Seyma Dumur, Ugurcan Sayili, Hafize Uzun

**Affiliations:** 1Department of Pediatrics, Medicine Hospital, Faculty of Medicine, Istanbul Atlas University, 34403 Istanbul, Turkey; yusuf.elgormus@medicinehospital.com.tr (Y.E.); dmemhs@gmail.com (O.O.); 2Department of Medical Biochemistry, Faculty of Medicine, Istanbul Atlas University, 34403 Istanbul, Turkey; seyma_dumur@hotmail.com; 3Department of Public Health, Cerrahpasa Faculty of Medicine, Istanbul University-Cerrahpasa, 34320 Istanbul, Turkey; drugurcansayili@gmail.com

**Keywords:** children, risk factors, vitamin B12, homocysteine, folic acid, ferritin

## Abstract

*Background*: Vitamin B12 is a water-soluble vitamin with important cellular functions; it is an essential vitamin. The aim of this study is to determine the B12 levels of children in the period from the 6th month when they start taking additional foods to the age of seven (preschool children) and the risk factors affecting them. *Methods*: One hundred pediatric patients aged 6–72 months who were diagnosed with vitamin B12 deficiency and their parents who agreed to attend Istanbul Atlas University, Medical Faculty, “Medicine Hospital” Pediatric Clinic between September 2022 and June 2023 were prospectively included in this study. *Results*: B12 deficiency was significantly higher in the 6–11 (25%)-month group than in the 12–23 (5.8%)- and 24–47 (2.8%)-month groups. Homocysteine levels were highest in those with insufficient B12 levels compared to the other groups. There was no statistically significant difference in weekly dairy and meat consumption levels between age groups. B12 levels were lower in the 6–11-month group than in the other groups. Homocysteine levels were highest in those with insufficient B12 levels (<200 pg/mL (148 pmol/L)). Folic acid levels were lower in the 24–47-month and 48–72-month groups than in the 6–11-month and 12–23-month groups. *Conclusions*: The results obtained in this study showed that low vitamin B12 and increased homocysteine levels seem to be important risk factors in preschool children, especially from the 6th month when they start consuming additional foods. The diagnosis of B12 deficiency can be confirmed by elevated serum total homocysteine levels, which are evidence of functional cobalamin deficiency.

## 1. Introduction

Vitamin B12 (B12; also known as cobalamin) is a water-soluble vitamin that cannot be synthesized in the human body and whose main source is animal foods such as meat, fish, and dairy products. Problems such as megaloblastic anemia and neuromotor developmental delay that occur in its deficiency are causes of serious morbidity [1,2]. B12 deficiency is an important health problem in Turkey as well as worldwide [3]. Epidemiological studies emphasize that the rate of cobalamin deficiency varies between 5% and 60%. It has been stated that this change also shows a correlation with age [4].

Vitamin B12 has important functions in the body. One of these functions is to use it as a cofactor while forming methionine from homocysteine. In the deficiency of this vitamin, there is an increase in homocysteine levels [5]. Vitamin B12 deficiency is mostly due to cobalamin absorption insufficiency, which occurs due to functional disorders in the gastrointestinal tract [6]. Hypotonia, apathy, adynamia, loss of visual contact, lethargy, and coma may be seen in children of mothers with low socioeconomic status or inadequate food intake or with accompanying signs of pernicious anemia [7].

Vitamin B12 stores of newborn babies depend on their mothers’ B12 levels. However, after the 6th month, with the intake of additional nutrients, external B12 intake begins. Since vitamin B12 is mainly found in animal foods, deficiency is common in children, especially in developing countries, in those who do not consume enough of this type of food. Since B12 deficiency was common in children who visited our outpatient clinic, we wanted to evaluate these patients in terms of their risk. The aim of this study is to determine the B12 levels of children in the period from the 6th month when they start taking additional foods to the age of seven and the risk factors affecting them.

## 2. Materials and Methods

This prospective study was approved by the Istanbul Atlas University ethics committee, Istanbul (approval date: 12 August 2022; number: E-22686390-050.99-19213) and was conducted in accordance with the Declaration of Helsinki. An informed consent form was obtained from the legal parents for each child included in the study. One hundred and sixty children aged 6–72 months who were diagnosed with vitamin B12 deficiency and their parents who agreed to attend Istanbul Atlas University, Medical Faculty, “Medicine Hospital” Pediatric Clinic between September 2022 and June 2023 were prospectively included in this study.

Detailed anamnesis (complaints on admission, whether there is a diagnosed disease or a drug that they use constantly, weekly dairy and meat consumption of the children, especially the vitamin supplements) was obtained and recorded on the forms specially prepared for the study. Detailed physical examinations were performed, and growth and development assessments, which may be clinical signs of B12 deficiency, were performed and recorded.

The social, economic, anthropometric and nutritional levels of the family (mother and father) were also questioned.

### 2.1. Exclusion Criteria

Children were excluded from the study if they were taking any medication, had a disease related to parasitosis and malabsorption, had acute infection, mental retardation, hereditary neuropathies, neurodegenerative diseases, autoimmune diseases, primary cardiac pathologies, and hormonal disorders, as well as if their parents were vegan or vegetarian. Our aim was to include in our study the children who switched to supplementary food and children who reached school age. For this reason, children who started school were excluded from the study.

### 2.2. Biochemical Analysis

Blood samples were taken in standardized tubes without anticoagulant and containing ethylenediamine tetraacetic acid (EDTA) for complete blood count (CBC) parameters. All biochemical parameters and CBC in the study were analyzed on the same day.

The CBC results were recorded with an automatic hematology analyzer (Sysmex XN-1000, Kobe, Japan). The neutrophil–lymphocyte ratio (NLR) and the platelet–lymphocyte ratio (PLR) were calculated from neutrophil/lymphocyte/thrombocyte counts.

Serum vitamin B12, ferritin, homocysteine, and folate levels were measured by Chemiluminiscent Microparticle Immuno Assay (CMIA) technology with assay protocols, referred to as Chemiflex. The required kits were purchased from Abbott Diagnostics. All the parameters were analyzed on an automated biochemistry analyzer (Architect i2000, Abbott Park, IL, USA). Vitamin B12 levels < 200 pg/mL (148 pmol/L) were defined as indicative of deficiency and 200–300 pg/mL (148–221 pmol/L) was defined as a borderline value [1,8].

### 2.3. Statistical Analysis

Statistical Package for the Social Sciences version 21.0 software package for Windows (IBM Corp., Armonk, NY, USA) was used for data evaluation and analysis. Jamovi 2.3.18 was used to create figures. Categorical variables are presented as frequencies (n) and percentages (%), and numerical variables are presented as medians (25. percentile–75th percentile). The Kolmogorov—Smirnov test was applied for normality analysis. The chi-square test or Fisher’s exact test was used to compare the distribution of categorical variables between groups. One-way ANOVA or the Kruskal—Wallis test were used to compare continuous variables between three independent groups (vitamin B12 level groups). The Tukey HSD test was used for the post hoc analysis after one-way ANOVA; adjusted *p* values were used after the Kruskal—Wallis test. A value of *p* < 0.05 was accepted as statistically significant.

## 3. Results

The characteristics of the participants by age group are shown in Table 1.

Of those included in the study, 55.9% were male, and 93.5% were term. A total of 26.5% used only breastmilk for the first 6 months. Vitamin B12 levels were <200 in 11.2%, 200–299 in 23.5%, and ≥300 in 65.3%.

Age groups, gender, term status, nutritional status in the first 6 months, and mother or father education status were similar. At 24 months and later, the use of vitamin supplements decreased significantly (*p*: < 0.001), while fish oil and vitamin B supplementation did not show a statistically significant difference; vitamin D and iron supplementation decreased with the increasing age. A total of 25% of those in the 6–11-month group had vitamin B12 deficiency; 5.8% of those in the 12–23-month group, 2.8% of those in the 24–47-month group, and 8.8% of those in the ≥48-month group had B12 deficiency. B12 deficiency was significantly higher in the ≤11-month group than in the 12–23- and 24–47-month groups (*p* < 0.001).

In Table 2, demographic, food consumption and laboratory characteristics by age groups are shown. B12 levels were significantly lower in the 6–11-month group (282; 201.5–405.5) than in the other groups (*p* = 0.001). Homocysteine levels were significantly higher in the 11-month group (7.38; 5.62–9.4) than in the 24–47-month group (*p* = 0.002). Folic acid levels were significantly lower in the 24–47-month (10.34 ± 3.75) and ≥48-month (9.35 ± 3.93) groups than in the ≤11-month (14.86 ± 2.42) and 12–23-month (14.09 ± 3.55) groups (*p*: <0.001). Hgb levels were significantly lower in the ≤11-month group (11.35 ± 0.93) and 12–23-month group (11.68 ± 1) than in the ≥48-month group (12.23 ± 0.97) (*p* = 0.001). NLR, PLR and SII levels were not significantly different between age groups (*p*: 0.638; 0.746; 0.620).

Table 3 shows the distribution of demographic and nutritional characteristics according to B12 levels. A total of 57.9% of those with insufficient B12 levels (<200), 57.5% of those with borderline B12 levels (200–299), and 55% of those with normal B12 levels (≥300) were male; B12 levels and gender distribution were similar (*p* = 0.972). The distribution between B12 levels and term delivery was similar. A total of 63.2% of those with insufficient B12 levels, 40% of those with borderline B12 levels, and 18% of those with normal B12 levels were ≤11 months; insufficient and borderline B12 levels at ≤11 months were significantly higher (*p* < 0.001). The rate of feeding with formula/supplementary food was significantly higher in patients with normal B12 levels (*p* = 0.026). There was no significant relationship between B12 levels and vitamin supplementation and its subtypes. No significant relationship was found between maternal or paternal education and B12 levels.

Table 4 shows the relationship between B12 levels and demographic, nutritional and laboratory characteristics. The weekly consumption of dairy products was significantly lower in those with insufficient B12 levels than in those with normal levels. There was a significant difference in homocysteine between all three B12 levels; homocysteine levels were higher in those with insufficient B12 levels (*p* < 0.001). There was no statistically significant difference between B12 levels in terms of ferritin, NLR, PLR, and SII.

While B12 levels and homocysteine levels showed a negative correlation (r = −0.63) in the whole group, no significant correlation was observed in those with insufficient and borderline B12 levels; however, a negative correlation (r = −0.320) was found in those with normal B12 levels. While there was no significant correlation between folic acid levels and B12 levels in the whole group, in those with insufficient and borderline B12 levels, a positive correlation (r = 0.270) was observed in those with normal B12 levels. The correlation between B12 levels and homocysteine in all groups is shown in Figure 1.

## 4. Discussion

There is no clear consensus on blood concentrations of B12 in the world [9]. In the present study, B12 deficiency was significantly higher in the 6–11 (25%)-month group than in the 12–23 (5.8%)- and 24–47 (2.8%)-month groups. Homocysteine levels were highest in those with insufficient B12 levels compared to those in the other groups. There was no statistically significant difference in weekly dairy and meat consumption levels between age groups. Another important result obtained was that homocysteine levels were highest in those with insufficient B12 levels (<200 pg/mL (148 pmol/L)). Folic acid levels were lower in the 24–47-month and 48–72-month groups than in the 6–11-month and 12–23-month groups. Vitamin B12 is important for healthy body functions because it has important cellular functions. It has an active role in growth, development and neurologic processes, especially in children. Therefore, children are highly susceptible to B12 deficiency. The results are important for identifying the risk group and taking precautions. Evaluation of blood homocysteine levels in terms of risk, especially when breastfeeding is discontinued and supplementary food is started, is important in terms of prevention of future disorders.

When the causes of micronutrient deficiencies are investigated, poor socioeconomic conditions are frequently mentioned. To determine etiology, we questioned the income status of the family, educational status and occupation of the parents, vitamin supplementation and dietary patterns of the children. In the current study, no statistically significant relationship was found between B12 deficiency and the educational level and occupation of the mother and father. At 24 months and later, the intake of vitamin supplements decreased significantly, while fish oil and vitamin B supplementation did not show a statistically significant difference. Supplementary intake, especially that of vitamin D and iron supplementation, decreased with the increasing age. B12 deficiency (<200 pg/mL) was seen in 12 patients aged 6–11 months, 3 patients aged 12–23 months, and 1 patient aged 24–47 months, and no B12 deficiency was seen in patients aged 48–72 months despite not receiving vitamin D and iron. B12 at the limit (200–299 pg/mL) was seen in 16 subjects aged 6–11 months, 15 subjects aged 12–23 months, 6 subjects aged 24–47 months and 3 subjects aged 48–72 months. On the whole, if RDW is increased with anemia in the blood count, nutritional deficiency should be considered; if MCV is low, iron deficiency should be considered. B12 deficiency or folic acid deficiency if MCV is high may be in question. Also, in the current study, anemia due to erythrocyte morphology and pathophysiology, vitamin B12 and folic acid deficiency was not observed. This is because vitamin B12 can be stored in the liver. Normally, a child is born with enough vitamin B12 stores until the end of their first year.

Studies including iron, vitamin B12 and folate deficiencies according to age groups have been conducted. When the participants were categorized according to B12 levels, although there were differences in HGB, HCT, MCV and MCH levels, all values were within normal values in current study. The values of the new inflammatory markers NLR, PLR and SII were at similar levels. In Atlanta, Ramussen et al. [1] found that vitamin B12 deficiency was high in the 12–18-year-old group. However, when clinicians recognize the symptoms of B12 deficiency, B12 levels should be checked even in the absence of anemia. It has been emphasized that neurological findings may be observed in patients with B12 and folate deficiency without megaloblastic anemia, and the neurological symptoms may be more severe [10,11,12]. In the last third of the 20th century, these deeply held misconceptions were slowly eroded with the application of folate and vitamin B12 assays and other techniques to neuropsychiatric patients with and without megaloblastic anemia [10].

The incidence of vitamin B12 deficiency varies from society to society, different age groups, socioeconomic level and nutritional habits. There are a limited number of population-based studies on the prevalence of vitamin B12 in the world. Based on the limited data in the literature, it is estimated that the deficiency of this vitamin is large enough to be a public health problem in the world. The importance of determining the frequency of vitamin B12 deficiency is that the permanent effects of megaloblastic anemia and the negative consequences that may occur in neurological systems can be prevented with early diagnosis and treatment. Several countries, Turkey in particular, have reported or measured the prevalence of vitamin B12 deficiency or marginal vitamin B12 deficiency [3,9,13,14,15,16]. In Şanlıurfa, the eastern province of Turkey, vitamin B12 deficiency was detected at very high rates of 72% in pregnant mothers and 41% in the cord blood of babies [3]. Our study does not feature a screening study in this sense [3]. Vitamin B12 deficiency is also common in Turkey. Nutritional vitamin B12 deficiency is common in southwest of Turkey where a Mediterranean-style diet is commonly used. Response to treatment is rapid and effective. However, delays may result in neurological sequelae.

Although deficiencies, especially in infancy, are generally reported as case reports in Western studies, it has been reported that nutritional deficiency is more common in Turkey, India and Lebanon [17,18]. Our aim is to include children who switch to supplementary food and children who have reached school age in our study. In the current study, B12 < 200 pg/mL in 6 (31.6%), B12 = 200–299 pg/mL in 10 (25.0%) and B12 ≥ 300 pg/mL in 29 (26.1%) breastfed children. B12 < 200 pg/mL was not found in any of the children fed with formula and/or supplementary food, B12 = 200–299 pg/mL in 3 (7.5%) children and B12 ≥ 300 pg/mL in 29 (26.1%) children. All our data show the extent of vitamin B12 deficiency. Some authors state that the sensitivity of 200 pg/mL, which is considered the threshold value for serum vitamin B12 level, is low, even in individuals with serum concentrations of up to 350 pg/mL reported that they had symptoms [19,20,21]. If the value of 350 pg/mL is accepted as a reference, the number of patients with vitamin B12 deficiency increases much more significantly. In the studies of Savage et al. [22], it was stated that a 10–26% misdiagnosis could be made with B12 level alone, and the sensitivity could reach 99.8% by studying methylmalonic acid and homocysteine. Demonstration of vitamin B12 deficiency at the metabolic level is possible by measuring homocysteine and methyl malonyl acid. Similar to our study, in the study conducted by Minet et al. [23], vitamin B12 levels were found to be lower in children who received breast milk as the main nutrition and who were not switched to supplementary food on time. In addition, vitamin B12 deficiency was not found in children fed with formula. This was attributed to the fact that the low level of vitamin B12 (0.3–0.5 mcg/100 mL) in formula foods prevented vitamin B12 deficiency and the socioeconomic level of these families was higher. Vitamin B12 deficiency, which has an important role in brain development from the period in the womb, is a preventable cause of neurological deficits. Therefore, screening and treatment before clinical symptoms occur are very important. We think that our study is useful in this regard.

Vitamin B12 is a cofactor involved in DNA synthesis, methylation, and neurotransmitter synthesis and is also involved in the homocysteine/methionine cycle. Therefore, vitamin B12 deficiency during periods of rapid growth, such as neonatal and infancy, leads to much more important symptoms than the symptoms of anemia seen at other times. Two cobalamin-dependent reactions reduce plasma levels of two toxic materials. (1) Homocysteine is associated with vascular endothelial damage. (2) Methylmalonyl CoA can cause metabolic acidosis. In 2003, Monsen et al. [24] investigated the relationship between plasma cobalamin, folic acid, methylmalonic acid and total homocysteine in 700 children of different age groups between 4 days and 19 years. They found a strong correlation between cobalamin deficiency in the first 6 months of life and elevated serum homocysteine. In our study, B12 levels were significantly lowest in the 6–11-month group, and a significant difference was found between the groups. Although homocysteine and folic acid levels were within normal limits in all groups, the highest homocysteine level was significant in the 6–11-month group. A significant difference was found between the groups. In the >300 pg/mL group, there was a weak correlation between B12 and homocysteine and a positive correlation with folic acid. However, no correlation was found in the 6–11-month group. When all participants were evaluated, a highly negative correlation was found between B12 and homocysteine. Although homocysteine levels were highest in children with B12 < 200 pg/mL, no significant correlation was found. The reason for the lack of correlation may be the small number of children in our study. In a study conducted by Esnafoglu and Ozturan in Turkey, vitamin B12 deficiency or insufficiency and elevated homocysteine may have contributed to the etiopathogenesis of depression in children and adolescents with depression [25]. The homocysteine and B12 levels were found to be significantly lower in children with autism spectrum disorder [26,27]. Carbamazepine monotherapy may cause a significant increase in the levels of homocysteine and a significant decrease in the levels of serum folate and vitamin B12 in children with epilepsy, with significant changes in the abovementioned parameters occurring early in the course of treatment [28].

Reduced homocysteine, pyridoxine, folate and vitamin B12 levels could be a risk factor in the etiology of attention deficit hyperactivity disorders [29]. Verhoef et al. [30] reported that plasma folate and, to a lower extent, plasma vitamin B12 and plasma homocysteine concentrations were inversely related and that folate levels were the most important marker for plasma homocysteine, including individuals receiving adequate amounts of vitamins through diet. The prognosis of vitamin B12 deficiency depends on the severity and duration of deficiency. Therefore, the diagnosis should be made as early as possible, and treatment should be started [31,32,33]. It is important to consider, diagnose and treat vitamin B12 deficiency in childhood. Although the cost of treatment is quite low, delay in treatment can cause serious complications such as profound anemia and irreversible neurological damage. There is a strong correlation between plasma homocysteine levels and serum B12 levels in the first years of life, and the mean homocysteine level has been reported to be 6–9 µmol/L in healthy newborns and infants [34]. Karademir et al. [35] showed that homocysteine levels decreased below 7.5 mmol/L and that methyl malonic acid (MMA) excretion in urine remained within normal values when serum B12 levels were above 200 pg/mL in infants receiving only breast milk. Önal et al. [36] classified 250 newborn babies as those with serum homocysteine levels above and below 10 mmol/L and compared them with ROC curves in terms of serum B12 levels and found that the limit B12 level was 200 pg/mL. The prognosis of vitamin B12 deficiency depends on the severity and duration of the deficiency. Therefore, diagnosis should be made as early as possible and treatment should be started.

In addition, the fact that B12 deficiency is easy to diagnose and its treatment is cheap and effective provides a great advantage when considering the possible problems that may occur. Greater consideration of vitamin B12 deficiency in routine clinical practice would be appropriate for early diagnosis and treatment. It was observed that the deficiency is especially high during the period when supplementary food is introduced. The results are important for identifying the risk group and taking precautions. However, multicenter prospective studies are needed to determine the prevalence and causes of deficiency in a wider geography. We can say that checking serum homocysteine levels should be considered a more accurate approach when evaluating serum vitamin B12 levels. Recalling that the majority of the mechanisms that lead to deficiency are due to malabsorption, vitamin B12 replacement therapy application is recommended to be administered parenterally in order to be more beneficial, accompanied by a treatment regimen consisting of stable metabolites.

The most important limitation of the study is that the B12 levels of the mothers could not be analyzed. The second important problem is that serum MMA levels were not analyzed. A third important reason was the small number of participants in the study, especially those with vitamin B12 deficiency.

The results obtained in this study showed that low vitamin B12 and increased homocysteine levels seem to be important risk factors in children, especially from the 6th month when they start consuming additional foods. The diagnosis of B12 deficiency can be confirmed by elevated serum total homocysteine levels, which are evidence of functional cobalamin deficiency. Vitamin B12 deficiency and increased homocysteine levels should be considered in patients with neurological complaints, even if hematological findings are normal. In newborn babies, the assessment of vitamin B12 levels should be performed as a screening test; if it cannot be performed, at least one dose of vitamin B12 should be administered intramuscularly.

## Figures and Tables

**Figure 1 medicina-59-01809-f001:**
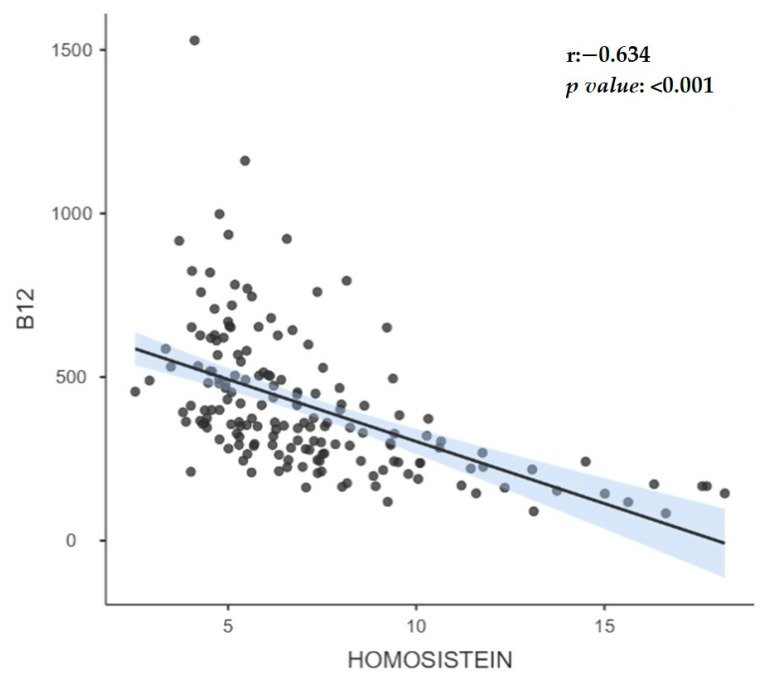
The correlation between B12 levels and homocysteine in all groups.

**Table 1 medicina-59-01809-t001:** The general characteristics of the participants by age groups.

	All Groups		6–11		12–23		24–47		≥48		
	n	%	n	%	n	%	n	%	n	%	*p*
**Gender**											
Boys	95 (55.9%)		26 (54.2%)		28 (53.8%)		19 (52.8%)		22 (64.7%)		0.715 *
Girls	75 (44.1%)		22 (45.8%)		24 (46.2%)		17 (47.2%)		12 (35.3%)		
**Term Status**											
Term	159 (93.5%)		46 (95.8%)		48 (92.3%)		34 (94.4%)		31 (91.2%)		0.881 ^†^
Preterm	11 (6.5%)		2 (4.2%)		4 (7.7%)		2 (5.6%)		3 (8.8%)		
**Nutrition Status**											
Breastfed	45 (26.5%)		12 (25.0%)		16 (30.8%)		10 (27.8%)		7 (20.6%)		
Breast milk + formula	35 (20.6%)		15 (31.3%)		8 (15.4%)		6 (16.7%)		6 (17.6%)		0.173 ^†^
Breast milk + supplementary food	58 (34.1%)		15 (31.3%)		22 (42.3%)		12 (33.3%)		9 (26.5%)		
Formula and/or supplementary food	32 (18.8%)		6 (12.5%)		6 (11.5%)		8 (22.2%)		12 (35.3%)		
**Vitamin supplement**	30 (17.6%)		12 (25.0%) ^a^		16 (30.8%) ^a^		1 (2.8%) ^b^		1 (2.9%) ^b^		<0.001 ^†^
Fish oil	2 (1.2%)		0 (0.0%)		0 (0.0%)		1 (2.8%)		1 (2.9%)		0.247 ^†^
Vitamin D	21 (12.4%)		11 (22.9%)		10 (19.2%)		0 (0.0%)		0 (0.0%)		<0.001 ^†^
Iron	15 (8.8%)		5 (10.4%)		10 (19.2%)		0 (0.0%)		0 (0.0%)		<0.001 ^†^
Vitamin B	2 (1.2%)		2 (4.2%)		0 (0.0%)		0 (0.0%)		0 (0.0%)		0.161 ^†^
**Education mother**											
Elementary education	27 (15.9%)		6 (12.5%)		5 (9.6%)		6 (16.7%)		10 (29.4%)		0.068 ^†^
High school/associate degree	72 (42.4%)		24 (50.0%)		17 (32.7%)		17 (47.2%)		14 (41.2%)		
Bachelor and above	71 (41.8%)		18 (37.5%)		30 (57.7%)		13 (36.1%)		10 (29.4%)		
**Education father**											
Elementary education	27 (15.9%)		6 (12.5%)		7 (13.5%)		8 (22.2%)		6 (17.6%)		0.721 ^†^
High school/associate degree	79 (46.5%)		24 (50.0%)		24 (46.2%)		13 (36.1%)		18 (52.9%)		
Bachelor and above	64 (37.6%)		18 (37.5%)		21 (40.4%)		15 (41.7%)		10 (29.4%)		
**B12 (pg/mL)**											
<200	19 (11.20%)		12 (25%) ^a^		3 (5.8%) ^b^		1 (2.8%) ^b^		3 (8.8%) ^a,b^		<0.001 ^†^
200–299	40 (23.50%)		16 (33%)		15 (28.8%)		6 (16.7%)		3 (8.8%)		
≥300	111 (65.30%)		20 (41.70%) ^a^		34 (65.4%) ^a,b^		29 (80.6%) ^b^		28 (82.4%) ^b^		
**B12 (pg/mL)**											
<300	59 (34.7%)		28 (58.3%) ^a^		18 (34.6%) ^a,b^		7 (19.4%) ^b^		6 (17.6%) ^b^		<0.001 *
≥300	111 (65.30%)		20 (41.70%) ^a^		34 (65.4%) ^a,b^		29 (80.6%) ^b^		28 (82.4%) ^b^		

*: chi-square test; ^†^: Fisher’s exact test. Different superscript letters indicate groups with significant differences.

**Table 2 medicina-59-01809-t002:** Demographic, food consumption and laboratory characteristics by age groups.

		All Groups	6–11	12–23	24–47	48–72	
			Mean ± Std	Mean ± Std	Mean ± Std	Mean ± Std	*p*
Age (Month)	Mean ± std	26.34 ± 21.61	8.58 ± 1.35	14.58 ± 3.54	31.47 ± 7.77	63.94 ± 11.98	-
	Median (Q1–Q3)	17 (10–36)	9 (8–10)	12 (12–17)	30.5 (24–36)	62 (53–72)	
Height (cm)	Mean ± std	86.05 ± 17.38	69.92 ± 3.94	78.14 ± 5.65	92.92 ± 8.14	113.66 ± 9.6	-
	Median (Q1–Q3)	79 (72.5–97)	70.25 (66–72.75)	78 (75–81)	94.5 (88.5–99)	113.65 (108–120)	
Weight (kg)	Mean ± std	12.84 ± 5.13	8.69 ± 1.09	10.85 ± 1.47	13.91 ± 2.42	20.58 ± 5.39	-
	Median (Q1–Q3)	11 (9.4–15.3)	8.6 (8–9.35)	10.78 (9.85–11.4)	14.05 (11.9–16)	19 (17–22.7)	
BMI (kg/m^2^)	Mean ± std	17.03 ± 2.24	17.81 ± 2.09	17.82 ± 2.04	16.07 ± 1.62	15.72 ± 2.35	-
	Median (Q1–Q3)	16.86 (15.72–18.01)	17.4 (16.46–19.07)	17.44 (16.4–19.1)	16.22 (14.92–16.85)	15.28 (14.31–16.58)	
Dairy products (day/week)	Mean ± std	5.86 ± 2.27	5.23 ± 2.87	5.88 ± 2.13	6.31 ± 2.01	6.26 ± 1.58	0.199 ^¶^
	Median (Q1–Q3)	7 (7–7)	7 (3–7)	7 (7–7)	7 (7–7)	7 (7–7)	
Meat consumption (day/week)	Mean ± std	2.31 ± 1.73	1.67 ± 1.56	2.71 ± 1.92	2.43 ± 1.59	2.49 ± 1.62	0.051 ^¶^
	Median (Q1–Q3)	2 (1–3)	2 (0–3)	2 (1.75–3.75)	2 (2–3)	2 (1.5–3)	
B12 (pg/mL)	Mean ± std	411.99 ± 215.58	322.23 ± 179.08	430.13 ± 233.84	491.11 ± 219.41	427.21 ± 193.51	0.001 ^¶^
	Median (Q1–Q3)	360.5 (264–506)	282 (201.5–405.5) ^a^	361.5 (272–532) ^b^	459.5 (332.5–635) ^b^	395 (317–514) ^b^	
Homocysteine (μmol/L)	Mean ± std	7.12 ± 3.03	8.46 ± 3.83	6.99 ± 2.91	5.75 ± 1.75	6.86 ± 2.23	0.002 ^¶^
	Median (Q1–Q3)	6.34 (5.01–8.16)	7.38 (5.62–9.4) ^a^	6.4 (5.03–8.26) ^a,b^	5.34 (4.64–6.49) ^b^	6.37 (5.33–7.52) ^a,b^	
Folic acid (ng/mL)	Mean ± std	12.57 ± 4.09	14.86 ± 2.42	14.09 ± 3.55	10.34 ± 3.75	9.35 ± 3.93	<0.001 ^T^
	Median (Q1–Q3)	13 (9.8–15.7)	14.95 (12.95–16.7) ^a^	14.3 (11.55–15.9) ^a^	10.35 (7–13.75) ^b^	8.3 (6.2–12.7) ^b^	
Ferritin (mL/ng)	Mean ± std	30.44 ± 24.13	36.99 ± 30.59	25.01 ± 19.36	27.85 ± 20.52	32.23 ± 22.45	0.181 ^¶^
	Median (Q1–Q3)	22.31 (14.21–40.24)	29.88 (15.28–48.15)	19.94 (12.03–32.18)	19.24 (13.29–43.44)	23.12 (19.09–39.54)	
Leukocyte (×10^3^/µL)	Mean ± std	10.36 ± 3.04	10.58 ± 3.1	11.13 ± 3.03	9.38 ± 2.52	9.9 ± 3.23	**0.037 ** ^¶^
	Median (Q1–Q3)	10.1 (8.31–12.14)	9.9 (8.91–11.8) ^a,b^	10.48 (9.14–12.75) ^a^	9.32 (7.41–10.82) ^b^	9.54 (7.75–12.14) ^a,b^	
Neutrophil (10^3^/µL)	Mean ± std	3.66 ± 1.06	3.78 ± 1.14	3.74 ± 0.99	3.64 ± 0.91	3.41 ± 1.17	0.432 ^T^
	Median (Q1–Q3)	3.7 (2.91–4.32)	3.86 (2.95–4.45)	3.9 (3.18–4.39)	3.69 (3.1–4.21)	3.17 (2.57–4.2)	
Lymphocyte (10^3^/µL)	Mean ± std	2.66 ± 1.14	2.6 ± 1.05	2.65 ± 1.25	2.51 ± 0.83	2.9 ± 1.34	0.831 ^¶^
	Median (Q1–Q3)	2.44 (1.9–3.2)	2.53 (1.8–3.2)	2.18 (1.89–3.35)	2.4 (2.09–2.97)	2.69 (2–3.9)	
HGB (g/dL)	Mean ± std	11.73 ± 1	11.35 ± 0.93	11.68 ± 1	11.83 ± 0.94	12.23 ± 0.97	**0.001 ** ^T^
	Median (Q1–Q3)	11.8 (11–12.4)	11.4 (10.6–12.05) ^a^	11.8 (11.05–12.3) ^a^	11.9 (11.35–12.3) ^a,b^	12.55 (11.4–12.8) ^b^	
HCT (%)	Mean ± std	35.06 ± 2.58	34.07 ± 2.47	35.04 ± 2.72	35.14 ± 2.18	36.39 ± 2.4	**0.001 ** ^T^
	Median (Q1–Q3)	34.95 (33.2–37)	33.8 (32.45–35.65) ^a^	34.8 (33.15–36.55) ^a,b^	34.9 (33.65–36.65) ^a,b^	37.1 (34.3–37.9) ^b^	
PLT (×10^3^/mL)	Mean ± std	344.52 ± 89.01	356.31 ± 101.39	343.58 ± 94.64	341.08 ± 81.79	332.94 ± 68.13	0.691 ^T^
	Median (Q1–Q3)	341.5 (288–399)	352 (295–406.5)	343 (273.5–402)	336 (289.5–380.5)	330 (290–370)	
MCV (fL)	Mean ± std	76.21 ± 4.76	75.59 ± 4.55	75.14 ± 4.99	77.17 ± 4.05	77.71 ± 5.01	**0.003 ** ^¶^
	Median (Q1–Q3)	76.9 (74.2–79.4)	76.15 (74.1–78.2) ^a^	76 (72.8–78.6) ^a^	77.5 (75.5–80.3) ^a,b^	78.65 (76.4–80.6) ^b^	
MCH (pg)	Mean ± std	25.92 ± 4.75	25.7 ± 4.4	25.9 ± 6.92	25.99 ± 2.84	26.18 ± 2.28	**0.013 ** ^¶^
	Median (Q1–Q3)	25.9 (24.3–27.1)	25.45 (24.15–26.5) ^a^	25.2 (24–27.05) ^a,b^	26.4 (25.4–27.35) ^a,b^	26.6 (25.4–27.5) ^b^	
MCHC (g/L)	Mean ± std	33.49 ± 1.41	33.31 ± 1.17	33.35 ± 1.59	33.64 ± 1.42	33.81 ± 1.38	0.307 ^¶^
	Median (Q1–Q3)	33.6 (32.7–34.5)	33.35 (32.5–34.05)	33.5 (32.3–34.4)	33.7 (33.05–34.65)	33.8 (33–34.8)	
NLR	Mean ± std	1.68 ± 1.06	1.69 ± 0.88	1.71 ± 0.86	1.8 ± 1.56	1.51 ± 0.93	0.638 ^¶^
	Median (Q1–Q3)	1.57 (1.05–2.06)	1.59 (1.13–2.05)	1.6 (1.14–2.08)	1.41 (1.07–1.98)	1.33 (0.71–2.11)	
PLR	Mean ± std	153.95 ± 80.22	160.13 ± 86.03	154.34 ± 76.78	157.89 ± 86.88	140.44 ± 70.99	0.746 ^¶^
	Median (Q1–Q3)	135.91 (101.44–181.48)	144.27 (108.24–185.09)	128.94 (94.75–188.33)	146.34 (104.96–183.55)	125.31 (82.86–174)	
SII	Mean ± std	573.85 ± 384.37	597.37 ± 375.9	580.78 ± 348.19	606.44 ± 502.3	495.55 ± 303.36	0.620 ^¶^
	Median (Q1–Q3)	507.82 (326.63–696)	547.07 (361–668.45)	520.04 (344.03–729.66)	506.7 (306.21–706.75)	419.08 (223.89–711.74)	

^¶^: Kruskal–Wallis test; ^T^: One-way ANOVA. BMI: Body mass index, HGB: hemoglobin, HCT: hematocrit, PLT: platelet, MCV: mean corpuscular volume; MCH: mean corpuscular hemoglobin, MCHC: mean corpuscular hemoglobin concentration, NLR: neutrophil-to-lymphocyte ratio, PLR: platelet-to-lymphocyte ratio, SII: systemic immune-inflammation index. Different superscript letters indicate groups with significant differences.

**Table 3 medicina-59-01809-t003:** The distribution of demographic and nutritional characteristics according to B12 levels.

	B12 Groups						
	<200		200–299		≥300		
	n	%	n	%	n	%	*p*
**Gender**							
Boys	11 (57.9%)		23 (57.5%)		61 (55.0%)		0.972 *
Girls	8 (42.1%)		17 (42.5%)		50 (45.0%)		
**Birth**							
Term	18 (94.7%)		38 (95.0%)		103 (92.8%)		1 ^†^
Preterm	1 (5.3%)		2 (5.0%)		8 (7.2%)		
**Age (month)**							
≤11	12 (63.2%) ^a^		16 (40.0%) ^a^		20 (18.0%) ^b^		
12–23	3 (15.8%)		15 (37.5%)		34 (30.6%)		<0.001 ^†^
24–47	1 (5.3%)		6 (15.0%)		29 (26.1%)		
≥48	3 (15.8%)		3 (7.5%)		28 (25.2%)		
**Nutrition**							
Breastfed	6 (31.6%)		10 (25.0%)		29 (26.1%)		
Breast milk + formula	5 (26.3%)		9 (22.5%)		21 (18.9%)		0.026 ^†^
Breast milk + supplementary food	8 (42.1%)		18 (45.0%)		32 (28.8%)		
Formula and/or supplementary food	0 (0.0%) ^a^		3 (7.5%) ^a^		29 (26.1%) ^b^		
**Vitamin Supplement**	5 (26.3%)		9 (22.5%)		16 (14.4%)		0.256 ^†^
Fish oil	0 (0.0%)		0 (0.0%)		2 (1.8%)		1 ^†^
Vitamin D	4 (21.1%)		6 (15.0%)		11 (9.9%)		0.304 ^†^
Iron	3 (15.8%)		5 (12.5%)		7 (6.3%)		0.181 ^†^
Vitamin B	0 (0.0%)		1 (2.5%)		1 (0.9%)		0.575 ^†^
**Education mother**							
Elementary education	1 (5.3%)		9 (22.5%)		17 (15.3%)		
High school/associate degree	7 (36.8%)		18 (45.0%)		47 (42.3%)		0.346 ^†^
Bachelor and above	11 (57.9%)		13 (32.5%)		47 (42.3%)		
**Education father**							
Elementary education	2 (10.5%)		6 (15.0%)		19 (17.1%)		
High school/associate degree	10 (52.6%)		22 (55.0%)		47 (42.3%)		0.675 ^†^
Bachelor and above	7 (36.8%)		12 (30.0%)		45 (40.5%)		

*: chi-square test; ^†^: Fisher’s exact test. Different superscript letters indicate groups with significant differences.

**Table 4 medicina-59-01809-t004:** The relationship between B12 levels and demographic, nutritional and laboratory characteristics.

	B12 Groups						
	<200 (pg/mL)		200–299 (pg/mL)		≥300 (pg/mL)		
	Mean ± Std	Median (Q1–Q3)	Mean ± Std	Median (Q1–Q3)	Mean ± Std	Median (Q1–Q3)	*p*
Age (month)	20.53 ± 24.2	9 (8–12) ^a^	19.58 ± 18.18	12.5 (9–22.5) ^a^	29.77 ± 21.68	24 (12–48) ^b^	<0.001 ^¶^
BMI (kg/m^2^)	17.86 ± 2.62	17.3 (15.49–20) ^a,b^	17.5 ± 1.87	17.29 (16.55–18.61) ^a^	16.71 ± 2.24	16.4 (15.28–17.75) ^b^	0.016 ^¶^
Dairy products (day/week)	4.74 ± 3.12	7 (1–7) ^a^	5.35 ± 2.53	7 (3–7) ^a,b^	6.24 ± 1.9	7 (7–7) ^b^	0.012 ^¶^
Meat consumption (day/week)	1.76 ± 1.77	2 (0–2.5)	2.39 ± 1.95	2 (1–3)	2.38 ± 1.64	2 (1.5–3)	0.273 ^¶^
Homocysteine (μmol/L)	12.6 ± 3.72	12.35 (8.92–16.32) ^a^	8.03 ± 2.36	7.46 (6.35–9.47) ^b^	5.85 ± 1.65	5.45 (4.64–6.84) ^c^	<0.001 ^¶^
Folic Acid (ng/mL)	14.53 ± 2.91	14.8 (13–17)	12.06 ± 4.12	11.9 (9.8–14.3)	12.41 ± 4.18	13.3 (9.2–15.8)	0.075 ^T^
Ferritin (mL/ng)	36.26 ± 36.05	22.4 (11.77–53.49)	27.43 ± 24.32	17.8 (10.3–36.05)	30.53 ± 21.52	23.15 (15.39–42.05)	0.296 ^¶^
Leukocyte (×10^3^/µL)	10.61 ± 4.28	9.45 (7.75–12.84)	10.53 ± 2.75	10.39 (8.33–11.93)	10.25 ± 2.91	10.07 (8.38–12.14)	0.787 ^¶^
Neutrophil (10^3^/µL)	3.44 ± 1.16	3.1 (2.41–4.5)	3.77 ± 1.13	3.86 (2.96–4.55)	3.67 ± 1.01	3.7 (3–4.2)	0.53 ^T^
Lymphocyte (10^3^/µL)	2.53 ± 0.91	2.3 (2–3.1)	2.66 ± 1.12	2.6 (1.95–3.2)	2.68 ± 1.18	2.37 (1.9–3.2)	0.982 ^¶^
HGB (g/dL)	11.56 ± 1.14	11.4 (10.6–12.5) ^a,b^	11.34 ± 0.9	11.4 (10.55–12) ^a^	11.9 ± 0.98	12 (11.3–12.5) ^b^	0.006 ^T^
HCT (%)	34.88 ± 2.86	35 (32.9–38) ^a,b^	34.15 ± 2.17	34.25 (32.55–35.35) ^a^	35.41 ± 2.61	35.4 (33.4–37.5) ^b^	0.027 ^T^
PLT (×10^3^/mL)	357.42 ± 99.37	349 (316–426)	350.58 ± 76.74	354.5 (295–406)	340.13 ± 91.69	335 (281–388)	0.655 ^T^
MCV (fL)	75.03 ± 5.2	76.4 (72.1–78.6) ^a,b^	75.05 ± 5.02	75.6 (72.95–78.35) ^a^	76.83 ± 4.51	77.3 (74.9–79.9) ^b^	0.03 ^¶^
MCH (pg)	24.66 ± 2.31	25 (22.7–26.5) ^a,b^	25.15 ± 2.62	25.3 (24.15–26.55) ^a^	26.41 ± 5.53	26.3 (25–27.2) ^b^	0.012 ^¶^
MCHC (g/L)	33.13 ± 1.16	33.3 (32.2–34.2) ^a,b^	33.18 ± 1.25	33.25 (32.4–34) ^a^	33.67 ± 1.48	33.7 (33–34.6) ^b^	0.045 ^¶^
NLR	1.52 ± 0.66	1.58 (1.07–1.88)	1.73 ± 0.98	1.6 (1–2.22)	1.7 ± 1.14	1.54 (1.03–2.06)	0.852 ^¶^
PLR	164.83 ± 85.8	152.17 (112.76–194.38)	156.09 ± 79.95	126.23 (105.06–194.11)	151.31 ± 79.92	135.5 (98.54–179.71)	0.733 ^¶^
SII	528.97 ± 267.95	446.81 (295.63–737.32)	611 ± 419.88	554.96 (329.17–691.75)	568.15 ± 389.67	506.45 (326.63–709.23)	0.841 ^¶^
Height (cm)	76.1 ± 13.8	73.5 (66–75) ^a^	80.74 ± 15.35	75.5 (70–89) ^b^	89.67 ± 17.59	85 (75–104) ^c^	<0.001 ^¶^
Weight (kg)	10.40 ± 3.43	9.5 (8.1–11.20) ^a^	11.73 ± 4.93	10.7 (8.33–13.45) ^b^	13.65 ± 5.26	11.8 (9.9–16.5) ^c^	<0.001 ^¶^

^¶^: Kruskal–Wallis test; ^T^: One-way ANOVA. Significant differences are shown with the letters ^a^, ^b^, ^c^ in the columns. BMI: Body mass index, HGB: hemoglobin, HCT: hematocrit, PLT: platelet, MCV: mean corpuscular volume; MCH: mean corpuscular hemoglobin, MCHC: mean corpuscular hemoglobin concentration, NLR: neutrophil-to-lymphocyte ratio, PLR: platelet-to-lymphocyte ratio, SII: systemic immune-inflammation index.

## Data Availability

Participant-level data are available from the corresponding author.

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
