# Peer review of "The Epidemiology of Deficiency of Vitamin B12 in Preschool Children in Turkey"

_medicina, 2023, doi:10.3390/medicina59101809_

Round 1

Reviewer 1 Report

Hello,

Thank you for giving me the opportunity to review this article.

It is a robust research paper with high-quality data.

However, I have some observations: - it is not clear to me why you chose 7- years as the upper limit of age in your patients. - review the Discussion section and focus on the clinical significance of your research data; - add a Conclusion section where you should emphasize the practical/ clinical consequence of your paper; - and improve/ extend the Bibliography section.

The connection between the deficiency of Vitamin B12 and some clinical diseases is not very clear, besides those mentioned in the literature.

As a pediatric surgeon, I would like to see more practical consequences of your paper. 

Looks OK, need to check again for some minor spelling issues.

Author Response

Comments and Suggestions for Authors

Thank you for giving me the opportunity to review this article.

It is a robust research paper with high-quality data.

However, I have some observations: - it is not clear to me why you chose 7- years as the upper limit of age in your patients. - review the Discussion section and focus on the clinical significance of your research data; - add a Conclusion section where you should emphasize the practical/ clinical consequence of your paper; - and improve/ extend the Bibliography section.

Our aim is to include children who switch to supplementary food and children who have reached school age in our study. For this reason, we excluded children who started school from the study.

We can say that checking serum homocysteine levels should be considered a more accurate approach when evaluating serum vitamin B12 levels. Recalling that the majority of the mechanisms that lead to deficiency are due to malabsorption, vitamin B12 replacement therapy application; In order to be more beneficial, we recommend that it be administered parenterally, accompanied by a treatment regimen consisting of stable metabolites.

The connection between the deficiency of Vitamin B12 and some clinical diseases is not very clear, besides those mentioned in the literature.

Necessary arrangements have been made in line with your request.

As a pediatric surgeon, I would like to see more practical consequences of your paper.

Necessary arrangements were made.

Comments on the Quality of English Language

Looks OK, need to check again for some minor spelling issues.

Reviewer 2 Report

Summary:

This research examines the epidemiology of Vitamin B12 deficiency in young children in Turkey from the age of 6 months, when they begin supplementary feeding, to 7 years. The study underscores the significance of low Vitamin B12 and increased homocysteine levels as noteworthy risk factors, particularly from the 6th month onward. The data suggests a clinical link between B12 deficiency and increased homocysteine levels, hinting at the potential implications of these deficiencies in the broader health context. Although the study is enlightening, there are sections, as highlighted in the review, where clarity and structure could be improved to improve the coherence and interpretability of the manuscript.

Comments:

Title

If this study is about Turkey or a specific region, authors should indicate this in the title, for example: "Epidemiology of Vitamin B12 Deficiency in Small Children in Turkey."

Introduction section

1. There is a jarring shift when the authors mention that " high B12 values suggest that leukemia may develop later in some pediatric patients". This point seems out of place and could be introduced more smoothly or contextualized within the scope of the article.

2. It would be interesting to know if the B12 deficiency is a more pronounced problem in Turkey compared to other countries, and if so, why. This could make the study more relevant.

Methods section

1.Authors must describe the storage and processing procedure for blood samples, if any.

2.When mentioning the comparison between three independent groups, what are these groups? Are these groups based on vitamin B12 levels or some other criterion?

Results section

1. At times, the data appears repetitive, particularly concerning B12 levels. Authors should present these results more concisely to make interpretation clearer.

2. It is mentioned that some children have B12 levels below 200, between 200-299, and above 300. Authors must also present the clinical significance of these values in the results. believe it's crucial to understand what exactly these numbers mean in health terms.

Discussion section

1. The flow of the discussion is somewhat fragmented. Information is presented in blocks, but the transition between different points could be better. Consider restructuring the discussion so there's a clear logical flow, starting with the main findings, then comparing with prior literature, and finally discussing clinical implications.

2.Some sentences and paragraphs are long and packed with information. Authors should simplify and break down into smaller sentences/paragraphs to improve clarity.

3. The text mentions that vitamin B12 deficiency and elevated homocysteine levels may contribute to the etiopathogenesis of depression in children and adolescents. Could you clarify this information? Is this based on the results of this study or prior literature?

Author Response

Comments and Suggestions for Authors

 Summary:

This research examines the epidemiology of Vitamin B12 deficiency in young children in Turkey from the age of 6 months, when they begin supplementary feeding, to 7 years. The study underscores the significance of low Vitamin B12 and increased homocysteine levels as noteworthy risk factors, particularly from the 6th month onward. The data suggests a clinical link between B12 deficiency and increased homocysteine levels, hinting at the potential implications of these deficiencies in the broader health context. Although the study is enlightening, there are sections, as highlighted in the review, where clarity and structure could be improved to improve the coherence and interpretability of the manuscript.

 Comments:

Title

If this study is about Turkey or a specific region, authors should indicate this in the title, for example: "Epidemiology of Vitamin B12 Deficiency in Small Children in Turkey."

The title of the article has been changed to "Epidemiology of Vitamin B12 Deficiency in Preschool Children in Turkey".

Introduction section

  1. There is a jarring shift when the authors mention that " high B12 values suggest that leukemia may develop later in some pediatric patients". This point seems out of place and could be introduced more smoothly or contextualized within the scope of the article.

" high B12 values suggest that leukemia may develop later in some pediatric patients"

removed

Epidemiological studies emphasize that the rate of cobalamin deficiency varies be-tween 5% and 60%. It has been stated that this change also shows a correlation with age (4).

It was brought.

  1. It would be interesting to know if the B12 deficiency is a more pronounced problem in Turkey compared to other countries, and if so, why. This could make the study more relevant.

Necessary arrangements were made.

 Methods section

1.Authors must describe the storage and processing procedure for blood samples, if any.

All biochemical parameters and CBC in the study were analysed on the same day.

2.When mentioning the comparison between three independent groups, what are these groups? Are these groups based on vitamin B12 levels or some other criterion?

Thank you to the reviewer for valuable comment. Yes, that is correct; Vitamin B12 levels. Corrected to clear up the misinformation

One-way ANOVA or the Kruskal‒Wallis test was used to compare continuous variables between three independent groups (Vitamin B12 level groups).

Results section

  1. At times, the data appears repetitive, particularly concerning B12 levels. Authors should present these results more concisely to make interpretation clearer.

Thank you to the reviewer for valuable comment. We have revised the results section to make it more concise.

  1. It is mentioned that some children have B12 levels below 200, between 200-299, and above 300. Authors must also present the clinical significance of these values in the results. believe it's crucial to understand what exactly these numbers mean in health terms.

Necessary arrangements were made.

Discussion section

  1. The flow of the discussion is somewhat fragmented. Information is presented in blocks, but the transition between different points could be better. Consider restructuring the discussion so there's a clear logical flow, starting with the main findings, then comparing with prior literature, and finally discussing clinical implications.

Necessary arrangements were made.

2.Some sentences and paragraphs are long and packed with information. Authors should simplify and break down into smaller sentences/paragraphs to improve clarity.

Necessary arrangements were made.

  1. The text mentions that vitamin B12 deficiency and elevated homocysteine levels may contribute to the etiopathogenesis of depression in children and adolescents. Could you clarify this information? Is this based on the results of this study or prior literature?

These are the results of a study conducted by Esnafoglu and Ozturan in Turkey.

Necessary arrangements were made.
